# Reduction of knee joint load suppresses cartilage degeneration, osteophyte formation, and synovitis in early-stage osteoarthritis using a post-traumatic rat model

Ikufumi Takahashi [1,2]*, Keisuke Takeda[1], Taro Matsuzaki [3], Hiroshi Kuroki[2], Masahiro Hoso[3]

1 Section of Rehabilitation, Kanazawa University Hospital, Kanazawa, Ishikawa, Japan, 2 Department of Motor Function Analysis, Human Health Sciences, Graduate School of Medicine, Kyoto University, Kyoto, Japan, 3 Division of Health Sciences, Graduate School of Medical Science, Kanazawa University, Kanazawa, Ishikawa, Japan

* t_ikuhumi@med.kanazawa-u.ac.jp

**Data Availability Statement:** All relevant data are within the manuscript and its Supporting Information files.

## Abstract

The purpose of this study was to clarify the histological effect of reducing the loading to knee on cartilage degeneration, osteophyte formation, and synovitis in early-stage osteoarthritis (OA) using a post-traumatic rat model. Ten male rats were randomly allocated into two experimental groups: OA induction by surgical destabilization of medial meniscus (DMM, OA group) and hindlimb suspension after OA induction by DMM (OAHS group). The articular cartilage, osteophyte formation, and synovial membrane in the medial tibiofemoral joint were analyzed histologically and histomorphometrically at 2 and 4 weeks after surgery. The histological scores and changes in articular cartilage and osteophyte formation were significantly milder and slower in the OAHS group than in the OA group. At 2 and 4 weeks, there were no significant differences in cartilage thickness and matrix staining intensity between both the groups, but chondrocytes density was significantly lower in the OA group. Synovitis was milder in OAHS group than in OA group at 2 weeks. Reducing knee joint loading inhibited histological OA changes in articular cartilage, osteophyte formation, and synovial inflammation. This result supports the latest clinical guidelines for OA treatment. Further studies using biochemical and mechanical analyses are necessary to elucidate the mechanism underlying delayed OA progression caused by joint-load reduction.

## Introduction

Mechanical stress, such as joint loading, is reported to influence osteoarthritis (OA) development [1, 2]. Moderate stress, such as physiological loading and exercise, are indispensable for metabolism of the articular cartilage and to help inhibit OA development [3, 4]. On the other hand, excessive mechanical stress has been reported to be the cause of the cartilage deterioration and OA onset [5, 6]. Hence, in the major international guidelines for OA treatment, such

**Funding:** This study was supported by a JSPS KAKENHI grant-in-aid for Young Scientists B (number: 17K13051) and for Early Career Scientists (number: 20K19444).

as the Osteoarthritis Research Society International (OARSI) issued in 2014 [7], the National Institute for Health and Care Excellence of the United Kingdom, National Health Service issued in 2014 [8] and those of the American Academy of Orthopedic Surgeons issued in 2013 [9], weight management and use of walking aids are recommended to avoid excessive loading on knee joints. In addition, the latest OARSI guidelines announced in 2019 strongly recommend weight management as the core treatment for all individuals regardless of comorbidity [10].

Therefore, clinical evidence has shown that reducing knee joint loading is effective and beneficial. However, little is known about the histological effects of reducing joint loading on articular cartilage. Focusing on this point, we previously analyzed the histological effect of reducing the loading on normal articular cartilage. We reported that reducing the loading on knee joints caused disuse atrophy of articular cartilage, including thinning of articular cartilage and decreased matrix staining [11, 12]. Moreover, we analyzed the histological effect of reduced loading on OA progression in a drug-induced rat model [13]. As a result, we revealed that OA progression was histologically suppressed by reducing the loading on knee joints in 2019 [13]. Although this finding is very useful, there were three major limitations. First, this OA model has no pathophysiological relationship with clinical OA because the model is drug induced and causes chondrocyte death directly and early [14, 15]. Second, only articular cartilage was analyzed. Recently, it has been reported that OA is a disease caused by the interaction of multiple joint components including articular cartilage, subchondral bone, and synovial membrane [15–18]. Third, only semiquantitative assessment by histological scores was performed to assess OA progression because it is difficult to evaluate the histomorphology due to the drug-induced model. We searched for previous studies that have clarified these three limitations but were unable to find any. Therefore, the histological and histomorphological effects of decreasing knee joint loading on articular cartilage, subchondral bone, osteophyte formation, and synovial membrane are not clear.

In view of the above, the purpose of this study is to overcome these three limitations and to clarify the effect of decreasing the load on the knee joint on OA progression of knee. Hence, we chose the surgical destabilization of medial meniscus (DMM) model as our rat OA model because surgically induced OA models have the advantage of similarity to post-traumatic OA [19], and the DMM model is the most widely used rodent OA model [20]. In addition, we analyzed histological changes in articular cartilage and multiple joint components, namely subchondral bone, osteophyte formation, and synovial membrane. Finally, we performed additional histomorphological analysis for cartilage thickness, intensity of matrix staining, chondrocyte density, osteophyte length, and synovial membrane thickness.

## Materials and methods

### Experimental animals and animal care

The study protocol was approved by the Animal Research Committee of the Graduate School of Medicine of Kanazawa University (Kanazawa, Japan; approval no. 204125) and conducted in accordance with the ARRIVE guidelines [21] and guidelines for the care and use of laboratory animals of Kanazawa University.

Twenty male Wistar rats (8 weeks old) were purchased from Japan SLC (Shizuoka, Japan) and housed under normal conditions for 5 weeks before the start of the experiments to acclimatize the animals to their new environment. One rat was housed per cage in a sanitary ventilated room under controlled temperature and humidity conditions and a 12-h light–dark cycle with ad libitum access to food and water. The experimental animals were monitored 2–3 times per week to control their health status, including general food and water intake, surgical

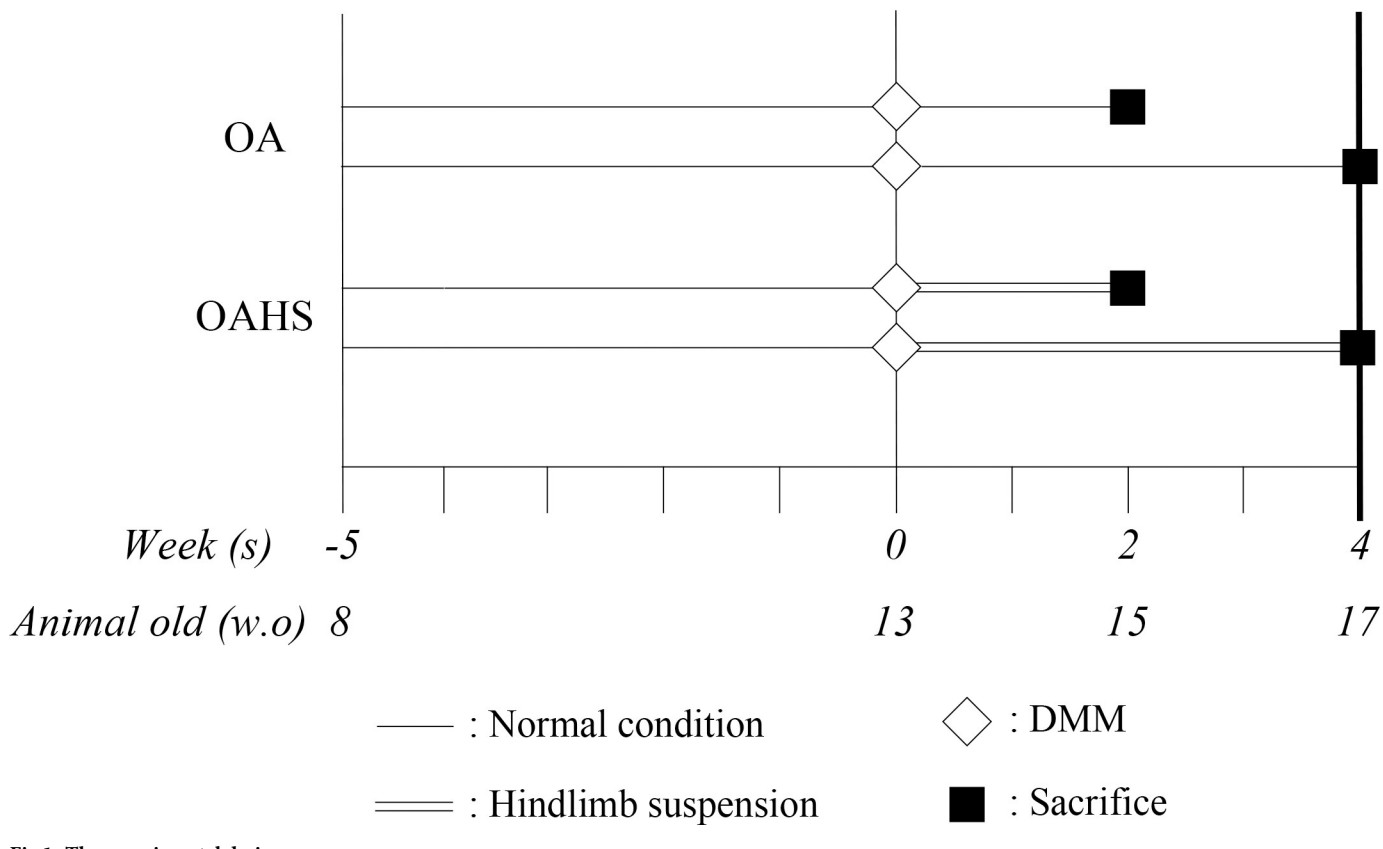

**Fig 1. The experimental design.**

wound condition, gait, and hindlimb suspension. The experimenter cleaned the cages once or twice every 2 weeks to keep the breeding environment clean.

The experimental protocol is shown in Fig 1. The rats were equally divided into two groups: post-traumatic OA (OA group) and hindlimb suspension after OA induction (OAHS group). OA was induced by surgical DMM. After surgical induction of OA, the rats were housed under normal conditions in the OA group and subjected to tail suspension during the experimental period in the OAHS group. The rats in the OAHS group were allowed to walk freely by using only their fore limbs. In both groups, we set the experimental period to 2 or 4 weeks after OA induction. Although it would have been ideal to evaluate OA until the end stage, long-term experimentation was difficult in the case of hindlimb suspension. Long-term suspension increases the suffering of the experimental animals and in some cases requires euthanasia. This also results in an increase in the number of experimental animals required. Consequently, this study focused only on early-stage OA.

After starting the experiment, no further interventions, including range of motion exercise, were performed during the experimental period. No analgesics or anti-inflammatory drugs were administered to any of the rats during the peri-operative period. The non-use of these drugs was approved by the Animal Research Committee of our university as described above and was following previous studies [3, 12, 22, 23].

The rats in the OA group (n = 10) and OAHS group (n = 10) were kept under normal conditions for 5 weeks. All rats in both groups underwent DMM surgery on the left knee and sham surgery on the right knee. The rats in the OA group were kept under normal conditions for 2 or 4 weeks (n = 5, respectively). The rats in the OAHS group were subjected to tail

suspension for 2 or 4 weeks (n = 5, respectively). After each experimental period, the rats were euthanized and assessed via histological analyses.

This study was concomitant to our previous study [12]. In order to reduce the number of animals used, based on the 3Rs principle of animal experiment (Reduction, Replacement and Refinement), a portion of the data used in this study was obtained from our previous study. Specifically, the data for the OA group at 2 and 4 weeks in the present study was the same as the data of the OA group at 2 and 4 weeks in the previous study. Since our previous study [12] is open-access and published under a CC-BY license, reuse of the data is permitted.

## Surgical induction of OA

The same highly experienced operators (IT and KT) performed all DMM surgeries [12]. DMM was induced by transecting the medial meniscotibial ligament (MMTL) in the left knee joint [22, 23]. In the OA and OAHS groups, for internal controls, a sham operation was performed on the right knee joint by using the same approach without MMTL transection. A previous study reported that there was no significant difference in OA severity between sham-operated and age-matched non-operated control joints [24].

## Hindlimb suspension

In the OAHS group, the rats were subjected to tail suspension throughout the experiment [11–13, 25]. Hindlimb suspension was performed according to Andries Ferreira's modified tail suspension method. Briefly, under inhalation anesthesia with isoflurane, the tail of each rat was disinfected. A sterile steel wire was then used to drill into the proximal coccyx in which the wire remained and shaped into a ring. The tail ring was then connected with another wire to a track hung above the cage, thereby enabling the animals to move freely on their forelimbs in the cage.

## Histological preparation

As described previously [12], decalcified paraffin sections were prepared for histology. Both knees were excised frontally to evaluate the histological changes in the medial tibiofemoral joints [11, 13]. Three paraffin sections (3-μm thickness) spaced at 200-μm intervals spanning the center of the medial tibiofemoral joint were stained with hematoxylin–eosin and toluidine blue to evaluate the severity and extent of cartilage lesions [26, 27]. Finally, the sections were viewed under a light microscope and imaged by using a digital camera (BX-51 and DP-74; Olympus Corporation, Tokyo, Japan) to evaluate the histological changes in the articular cartilage.

## Histological analyses for articular cartilage

To assess the histological changes of OA, we quantitatively evaluated the articular cartilage of the tibia in the medial tibiofemoral joint by using the OA cartilage histopathology assessment system [28]. The OARSI scoring system, consisting of six grades and four stages on a scale from 0 (normal) to 24 (severe cartilage lesion) was used for semiquantitative evaluation of cartilage lesion severity [28]. The maximum OARSI score was the highest score among 3 sections and provides a measure of the "severity" of the cartilage lesions [12, 26, 27]. The summed OARSI score was the total of the three section scores and provides a measure of the "extent" of the cartilage lesions [12, 26, 27].

All histological scores of these scoring systems were determined by two blinded and trained independent observers (MH, a pathologist and IT). In our previous study, interclass

correlation coefficients for the intra- and inter-rater reliabilities of the OARSI score with 95% confidence intervals were excellent: 0.94 (0.92–0.95) and 0.91 (0.89–0.93), respectively [13].

## Histological analyses for subchondral bone and synovial membrane

For evaluation of subchondral bone damage and synovial inflammation, we chose a single section with the highest OARSI score among the three sections. To quantitatively evaluate subchondral bone damage of the tibia in the medial tibiofemoral joint, we used the calcified cartilage and subchondral bone damage score [29]. This score was established by Gerwin et al. in 2010 and ranges from 0 to 5, with higher values indicating severe subchondral lesion (S1 Table. Subchondral bone damage score) [29]. In addition, to quantitatively evaluate synovial inflammation in the medial tibiofemoral joint, we used the scoring system for synovitis (SSS, S2 Table. Scoring system for synovitis) [30]. This system consists of three items of hyperplasia, infiltration, and stroma. Each of them is judged on a four-level scale, and the total score is calculated. Therefore, the score is 0–9, with a higher score indicating more severe synovial inflammation.

## Histomorphometrical analyses for articular cartilage

As described previously [11, 12, 28, 31, 32], Adobe Photoshop CC imaging software (Adobe Systems, Inc., San Jose, CA, USA) was used to perform histomorphometrical analyses for articular cartilage to evaluate the following four parameters: cartilage thickness, intensity of matrix staining with toluidine blue, chondrocyte density, and osteophyte length. For evaluation of cartilage thickness, staining intensity, and chondrocyte density, we chose the single section with the highest OARSI score among 3 sections. For evaluation of osteophyte length, we chose the single section with the longest osteophyte among the three sections.

To evaluate cartilage thickness, digitized images of the sections stained with toluidine blue were used. The cartilage thickness was defined as the distance between the cartilage surface and the osteochondral junction. We used the measured area of the cartilage with a width of 1 mm in the center of the lesion at the tibia in the medial tibiofemoral joint as the average cartilage thickness. To evaluate matrix intensity, digitized images of cartilage stained with toluidine blue were converted to greyscale (white, 255; black, 0) to assess the relative intensity of toluidine blue staining. The average staining intensity was calculated at the same area in the same manner as performed for the measurement of articular cartilage thickness. To evaluate chondrocyte density, digitized images of the sections stained with hematoxylin–eosin were used. Chondrocyte density was determined as the number of chondrocytes per unit area of cartilage. This unit area was calculated by using the same method as used for the above cartilage thickness, and the width to be measured was set to 200 μm. Chondrocytes with visible nuclei within the area of interest were counted manually.

## Histomorphometrical analyses for osteophyte and synovial membrane

For evaluation of osteophyte length, we chose a single section with the longest osteophyte among the three sections. To evaluate osteophyte formation of the tibia in the medial tibiofemoral joint, we used digitized images of sections stained with toluidine blue and measured the length from the deepest point of its base at the chondro-osseous junction to the surface of the overlying cartilage at its thickest point [29].

For evaluation of the inflammation of synovial membrane and measurement of osteophyte length, we chose a single section with the thickest synovial membrane among the three sections and measured the synovial thickness at the medial tibiofemoral joint.

## Statistical analysis

All statistical analyses were performed by using JMP 14 software (SAS Institute, Cary, NC, USA). All data were statistically analyzed as parametric data. The sample size was five for each group. Descriptive statistics were calculated as the median with interquartile range for the OARSI score, subchondral bone score, and synovitis score and as the mean with standard deviation for body weight and histomorphometrical data. We considered $P < .05$ as indicative of statistical significance for all analyses; exact $P$ values are shown in the figures. For all of the data, we performed analysis of variance followed by the post-hoc Tukey's honest significant difference test.

A sample-size calculation was performed by using the sample size and power tool in G\*power 3.1 (free; available at https://www.psychologie.hhu.de/arbeitsgruppen/allgemeine-psychologie-und-arbeitspsychologie/gpower.html) [33] and pilot experimental data of the main parameter and maximum and summed OARSI scores at 4 weeks, including the first five rats, in the OA and OAHS groups. For these scores, a minimum of 4 and 3 were required in the OA and OAHS groups, respectively, with a power of 0.95 and a significance level of $P < .05$. Therefore, we set the sample size to five rats per group.

## Results

### General condition

After a few minutes from the end of the inhalation-induced anesthesia, all rats attained consciousness and began to move around in their cages. No experimental animals died through accidents or other unexpected causes during any of the experiments. None of the rats developed infections in their knee joints or tail wounds. All animals completed the study safely, as per the experimental design, without any unexpected adverse events. Please refer to S3 Table (Body weight) for data on weight changes in the rats during the experiment.

### Histological scores for articular cartilage and subchondral bone

In Figs 2 and 3, the maximum and summed OARSI score was significantly lower in OAHS group than in the OA group at 2 and 4 weeks after OA induction. In the subchondral bone damage scores, no significant differences were found between the two groups at 2 and 4 weeks. Please refer to the S4 Table (Histological scores) for detailed information on the histological scores.

### Histomorphometrical analyses for articular cartilage and osteophyte

No significant differences in cartilage thickness in the operated limb were found between the OA and OAHS groups at 2 and 4 weeks after surgery. However, the cartilage thickness in the operated limb was significantly less than that in the sham limb in both groups at 2 and 4 weeks after surgery (Fig 4 and S5 Table. Histomorphometrical results).

No significant differences in matrix staining intensity by toluidine blue of the operated limb were found between the two groups at 2 and 4 weeks (Fig 4 and S5 Table. Histomorphometrical results).

Chondrocyte density of the operated limb was significantly lower in the OA group than that in the OAHS group at 2 and 4 weeks. And operated limb density was significantly less than sham limb density in the OA group at 2 and 4 weeks after surgery, but no significant differences were found between the operated and sham limbs in the OAHS group at 2 and 4 weeks (Fig 4 and S5 Table. Histomorphometrical results).

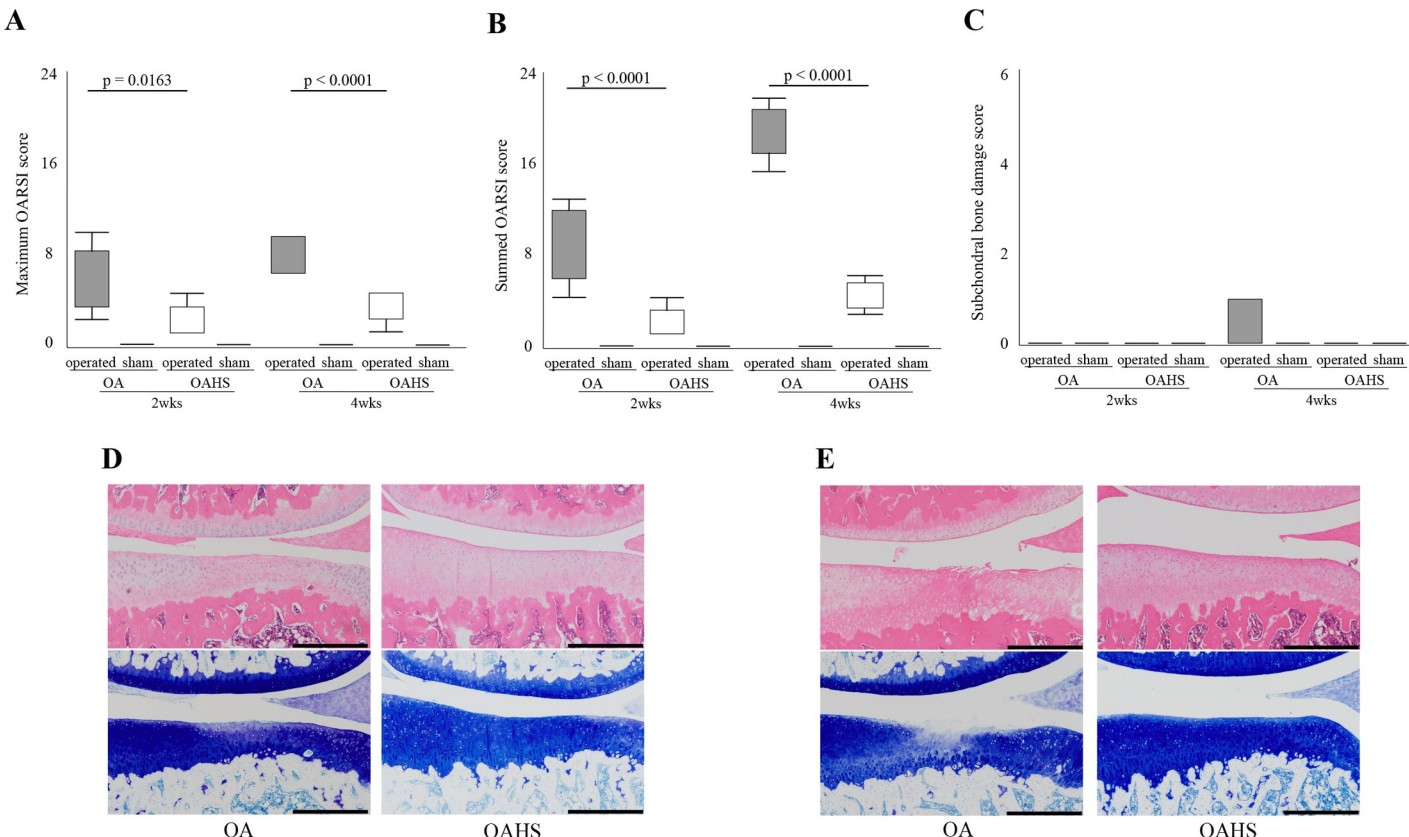

**Fig 2. Results of histological changes in the articular cartilage and subchondral bone.** The maximum OARSI score (A) and summed OARSI score (B) were significantly lower in the OAHS group than those in the OA group at 2 and 4 weeks. For the subchondral bone score (C), no significant difference was found between the two groups at 2 and 4 weeks. Representative histological changes in the articular cartilage are shown in (D) (2 weeks) and (E) (4 weeks). At 2 weeks, in the OA and OAHS groups, the articular surface was intact, but the staining intensity with toluidine blue was partially reduced in the OA group. At 4 weeks, fibrillation and fissures were detected and the area of reduced staining with toluidine blue was expanded in the OA group. In the OAHS group, a slight decrease in staining intensity on the surface of the articular cartilage was found, but the surface remained almost intact. Scale bar = 500 μm.

In the osteophyte length of the operated limb, no significant difference was found between the OA and OAHS groups at 2 weeks after surgery, but the length was longer in the OA group than in the OAHS group at 4 weeks. There was greater osteophyte formation in the operated limb than in the sham limb in both the groups at 2 and 4 weeks (Fig 5 and S5 Table. Histomorphometrical results).

### Histological score and histomorphometrical analyses for synovial membrane

In SSS, no significant differences in the operated limb were found between both groups at 2 and 4 weeks after surgery (Fig 6 and Supplementary Result). In synovial thickness, at 2 weeks after surgery, the thickness of operated limb was significantly thinner in OAHS group than in OA group; however, at 4 weeks, no significant difference in the operated limb was found between the OA and OAHS groups (Fig 6, S4 Table. Histological scores, and S5 Table. Histomorphometrical results).

## Discussion

The purpose of this study was to determine the histopathological effect of reduced knee joint loading on OA progression using surgical model that is pathologically relevant to human

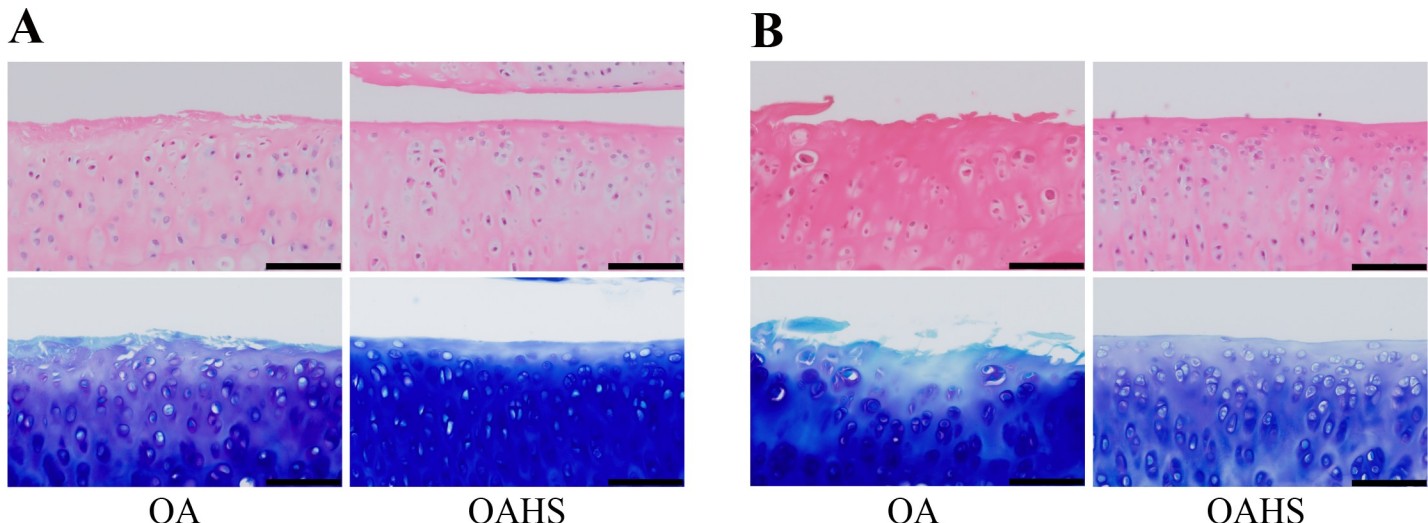

**Fig 3. Histological findings of the tibial articular cartilage at a high magnification.** (A) and (B) represent histological images at 2 and 4 weeks, respectively. In the OA group, fibrillation was detected at 2 weeks, and a moderate decrease in toluidine blue staining intensity was found from the surface to the middle layer. At 4 weeks, fibrillation and fissures were detected, and the staining intensity was further decreased. In the OAHS group, the articular surface was intact at both 2 and 4 weeks, but the overall staining intensity was slightly reduced. Scale bar = 100 μm.

secondary OA, and hindlimb suspension model. As a result, it was demonstrated that the histological OA changes in cartilage, osteophyte formation, and synovial inflammation developed more slowly when knee joint loading was reduced. Furthermore, reducing joint loading maintained matrix staining and chondrocyte density by inhibiting OA progression, thereby keeping the articular cartilage condition closer to normal histologically and histomorphologically. Therefore, these histological findings provide strong basic evidence to support the recommendation of weight management for OA in the above-mentioned guidelines [7–10].

In the present study, multiple joint components were evaluated, and the results showed no obvious significant difference in histological changes in the subchondral bone. Also, no significant difference in thickness and matrix staining of the articular cartilage was observed. This may be due to the fact that the evaluation period was only in the early postoperative period of 2–4 weeks. Ideally, the long-term course should be evaluated until the end of OA, such as 8–16 weeks, but the tail suspension for more than 8 weeks often leads to tail injury. For long-term

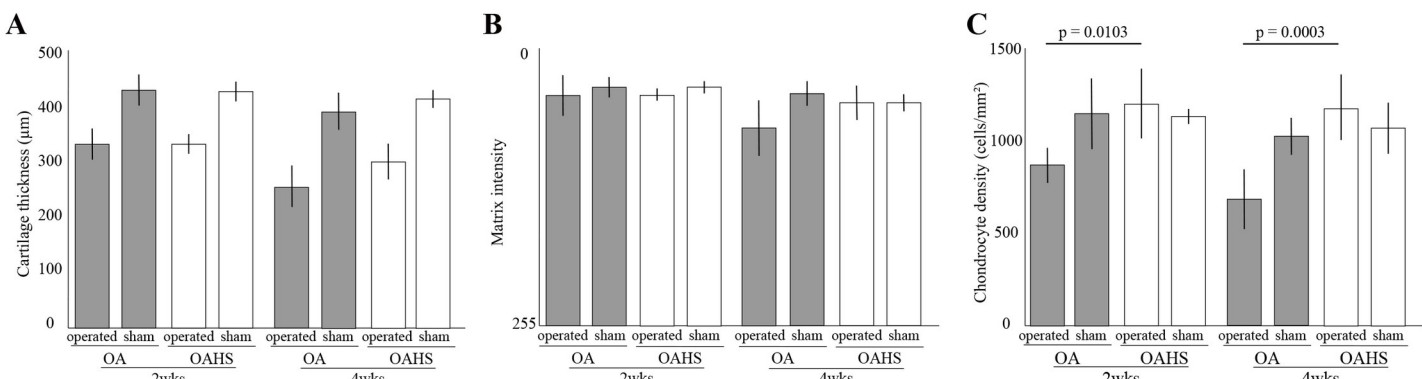

**Fig 4. Results of histomorphological changes in the articular cartilage.** (A-C) present cartilage thickness, staining intensity, and cell density, respectively. No significant differences in cartilage thickness and matrix intensity of the operated limb were found between the two groups at 2 and 4 weeks. Cell density of the operated limb in the OA group was significantly less than that in the OAHS group at 2 and 4 weeks.

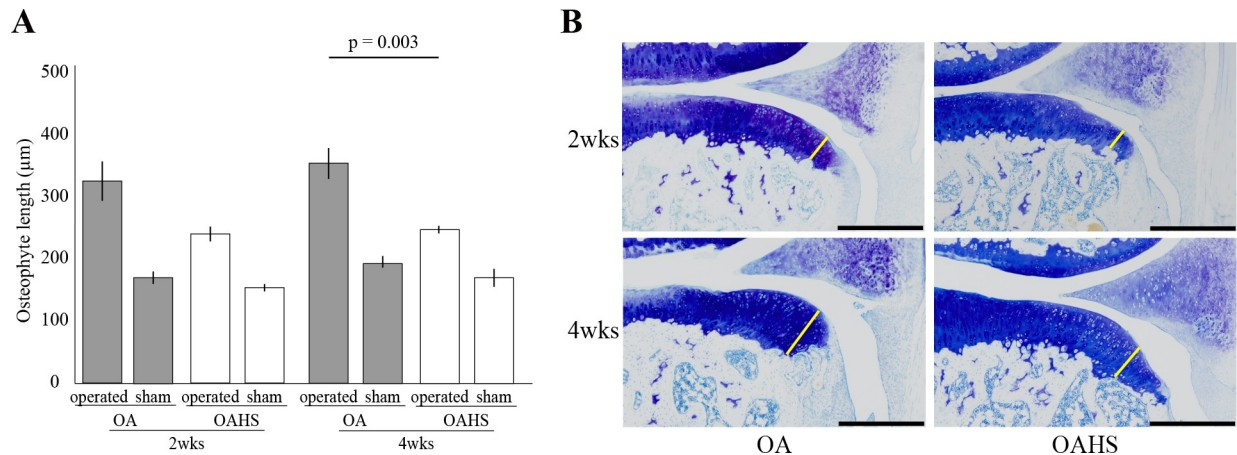

**Fig 5. Results of histological and histomorphological changes in osteophyte formation.** As the graph (A) and histological images (B) show, no significant difference in the operated limb was found between both groups at 2 weeks, but osteophyte length was longer in the OA group than that in the OAHS group at 4 weeks. Scale bar = 500 μm.

experiments in tail suspension, tail injury may cause distress to the experimental animal and dropout by euthanasia from the experimental design may be necessary. In such cases, more experimental animals would be needed, but such cases are inappropriate from the standpoint of experimental ethics. Therefore, our focus was to observe the early postoperative period of 2–4 weeks.

As shown by these histological and histomorphological results described above, OA progression was significantly suppressed by reducing the load on the knee joints. However, in hindlimb suspension models such as the one used in this study, disuse atrophy of the articular cartilage is a result of the unloading environment [4, 11, 12]. In the results of the present study, compared to the degree of OA progression, the OAHS group showed a decrease in cartilage thickness almost equal to that of the OA group. Therefore, we believe that it is highly likely that disuse atrophy of articular cartilage associated with non-loading has occurred. In addition,

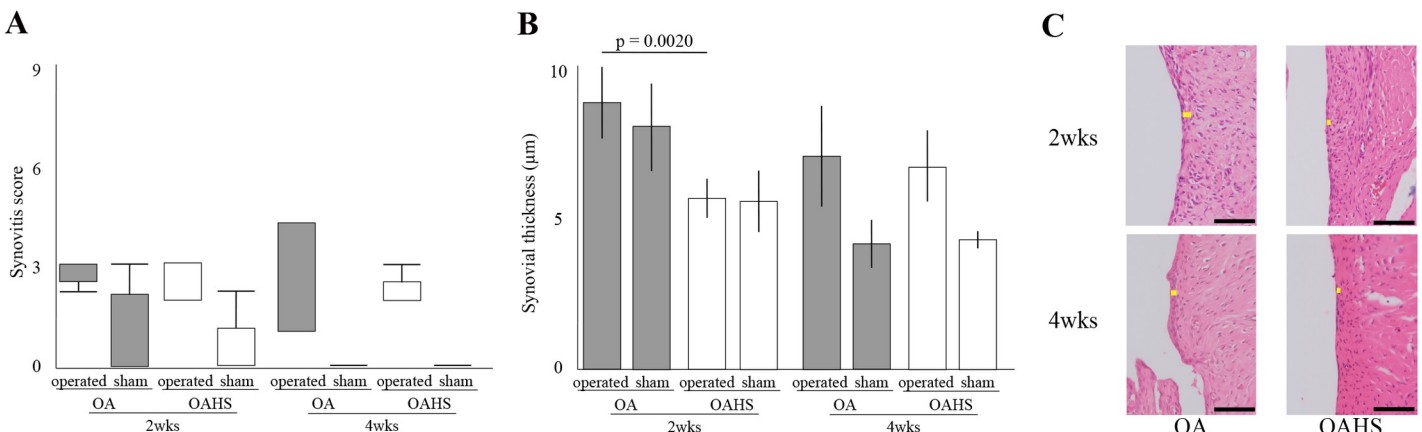

**Fig 6. Results of histological and histomorphological changes in synovial inflammation.** As the graph (A and B) and histological images (C) show, for SSS, no significant difference in the operated limb was found between the OA and OAHS groups throughout the experimental period. Regarding synovial thickness, at 2 weeks after surgery, the operated limb was significantly thinner in the OAHS group than that in the OA group. In the histological images, no significant difference in the operated limb was found between the OA and OAHS groups at 4 weeks. Slight hyperplasia of the synovial surface cells was observed in the OA group at 2 weeks. In both groups, slight inflammatory cell infiltration and slight stroma activation were observed at 2 and 4 weeks. Scale bar = 50 μm.

disuse atrophy of articular cartilage is one of the factors that accelerates OA progression, as we reported in our previous study [12]. Therefore, clinically, to suppress OA progression and not cause disuse atrophy of articular cartilage, intermittent loading or partial loading may be effective during long-term unloading. Further studies are necessary to clarify the appropriate amount and time of the loading.

The present study consists of the results of our previous study and additional experiments specified herein [12]. To avoid significant differences in experimental techniques between the two experiments, we paid careful attention to three aspects: surgery, staining, and evaluation. OA models were created by the same surgeon using the same surgical technique as in the previous study [22, 23]. Additionally, the surgeon's proficiency in the surgical technique was confirmed by preliminary experimentation. Furthermore, there was no marked difference in operative time between experiments. Staining was performed in the same environment and conditions, including staining solution used, staining time, room temperature, and laboratory equipments. Before conducting additional experiments, we preliminarily verified that there was no marked difference in staining intensity by the same technique. Histological evaluation was performed by two well-trained double-blinded experimenters. The same experimenter performed the histological evaluation in the two experiments and the two had high intra- and inter-rater reliability [13]. Based on the above, we consider our two experiments to have been conducted with high reliability and reproducibility.

This study had four limitations. First, only histological and histomorphometrical analyses were performed. To clarify the effect of load reduction on OA progression, further biochemical studies for catabolic markers, such as MMP-13 and ADAMTS5, or anabolic markers, such as aggrecan and type II collagen, would be necessary, and further studies using mechanical tests would also be needed. Second, the sample size was relatively small. We used G*power to calculate the required sample size, but due to the use of multiple comparisons in the statistical analysis and the large individual differences in scores, a larger sample size may be desired. Third, the sex of the experimental animals we used was male only; since OA generally occurs more frequently in females [34], it may have been necessary to equalize the sex of the experimental animals in this study. Fourth, there is no quantitative assessment of the mechanical stress occurring in the knee joint. Tail suspension is an experimental method that allows the rats to reduce bodyweight load on the knee joint. However, the rats can flex and extend their knee joints freely (actively or passively), and with this movement, mechanical stresses such as shear and compressive forces are generated in the articular cartilage. The metabolism of articular cartilage is influenced by these mechanical stresses [35, 36]. In addition, the number of steps and other activities of the experimental animals in their cages were not evaluated.

## Conclusions

Reducing knee joint loading inhibited histological changes in articular cartilage, osteophyte formation, and synovial inflammation in early-stage OA. In addition, reduced loading kept the articular cartilage condition closer to normal histologically and histomorphologically. Therefore, our findings support the latest clinical guidelines for OA treatment. Further studies using biochemical and mechanical analyses are required to clarify the structure underlying delayed OA progression caused by joint-load reduction.

## Supporting information

**S1 Table. Subchondral bone damage score.**
(DOCX)

**S2 Table. Scoring system for synovitis.**
(DOCX)

**S3 Table. Body weight.**
(DOCX)

**S4 Table. Histological scores.**
(DOCX)

**S5 Table. Histomorphometrical results.**
(DOCX)

## Acknowledgments

The authors thank the members of the Department of Human Pathology at the Kanazawa University Graduate School of Medicine for providing advice about the histopathological techniques.

## Author Contributions

**Conceptualization:** Ikufumi Takahashi, Keisuke Takeda, Hiroshi Kuroki, Masahiro Hoso.

**Data curation:** Ikufumi Takahashi, Keisuke Takeda.

**Formal analysis:** Ikufumi Takahashi, Keisuke Takeda, Taro Matsuzaki, Masahiro Hoso.

**Funding acquisition:** Ikufumi Takahashi, Taro Matsuzaki, Masahiro Hoso.

**Investigation:** Ikufumi Takahashi, Keisuke Takeda, Taro Matsuzaki, Hiroshi Kuroki, Masahiro Hoso.

**Methodology:** Ikufumi Takahashi, Keisuke Takeda, Taro Matsuzaki, Hiroshi Kuroki, Masahiro Hoso.

**Project administration:** Ikufumi Takahashi, Keisuke Takeda, Taro Matsuzaki.

**Resources:** Ikufumi Takahashi, Taro Matsuzaki, Masahiro Hoso.

**Software:** Ikufumi Takahashi.

**Supervision:** Ikufumi Takahashi, Keisuke Takeda, Taro Matsuzaki, Hiroshi Kuroki, Masahiro Hoso.

**Validation:** Ikufumi Takahashi, Taro Matsuzaki, Hiroshi Kuroki.

**Visualization:** Ikufumi Takahashi, Taro Matsuzaki, Hiroshi Kuroki.

**Writing – original draft:** Ikufumi Takahashi, Keisuke Takeda, Masahiro Hoso.

**Writing – review & editing:** Ikufumi Takahashi, Keisuke Takeda, Taro Matsuzaki, Hiroshi Kuroki, Masahiro Hoso.

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
