## [Decision Letter · Decision Letter 0]

15 Apr 2021

PONE-D-21-06339

Reduction of knee joint load suppresses cartilage degeneration, osteophyte formation, and synovitis in early stage using a post-traumatic rat model

PLOS ONE

Dear Dr. Takahashi,

Thank you for submitting your manuscript to PLOS ONE. After careful consideration, we feel that it has merit but does not fully meet PLOS ONE’s publication criteria as it currently stands. Therefore, we invite you to submit a revised version of the manuscript that addresses the points raised during the review process.

We look forward to receiving your revised manuscript.

Kind regards,

Lin Han

Academic Editor

PLOS ONE

Journal Requirements:

2. Thank you for submitting the above manuscript to PLOS ONE. During our internal evaluation of the manuscript, we found significant text overlap between your submission and the following previously published works, some of which you are an author.

https://journals.sagepub.com/doi/10.1177/1947603520982350

https://www.oarsijournal.com/article/S1063-4584(19)30896-9/fulltext

Please revise the manuscript to rephrase the duplicated text, cite your sources, and provide details as to how the current manuscript advances on previous work. Please note that further consideration is dependent on the submission of a manuscript that addresses these concerns about the overlap in text with published work.

3. As part of your revisions, please update your Methods/Results to address the following items pertaining to animal research, animal health and welfare.  (1) The frequency of animal monitoring, including the specific criteria you used to monitor animal health as well as a description of humane endpoints. (2) Specific animal welfare considerations, including efforts to alleviate suffering (supportive care - gel-packs, cage modifications/environmental enrichment, administration of analgesics, anesthetics, etc.), (3) The rate of mortality during the study (if applicable) and if animals died unexpectedly or prior to the experimental endpoint, an explanation for the cause of death. (4) Details of unanticipated adverse events such as illness or injury as a result of the experimental procedures; (5) Lastly, please complete and submit the ARRIVE Guidelines 2.0 checklist (Essential 10 version): https://arriveguidelines.org/resources/author-checklists.

Additional Editor Comments (if provided):

Reviewers' comments:

Reviewer's Responses to Questions

**Comments to the Author**

1. Is the manuscript technically sound, and do the data support the conclusions?

Reviewer #1: Yes

Reviewer #2: Partly

2. Has the statistical analysis been performed appropriately and rigorously? 

Reviewer #1: No

Reviewer #2: Yes

3. Have the authors made all data underlying the findings in their manuscript fully available?

Reviewer #1: Yes

Reviewer #2: Yes

4. Is the manuscript presented in an intelligible fashion and written in standard English?

Reviewer #1: Yes

Reviewer #2: Yes

5. Review Comments to the Author

Reviewer #1: In their manuscript “Reduction of knee joint load suppresses cartilage degeneration, osteophyte formation, and synovitis in early stage using a post-traumatic rat model” Takahashi et al. report the effect of reducing the loading on knee joints during early osteoarthritis (OA) progression by using rat OA model. They find that the histological changes in articular cartilage and osteophyte formation were milder and slower with decreasing load to the knee. With more loading, the chondrocyte density is significantly lower. Synovitis was severe with loading increase. In summary, with reducing knee joint loading, the histological OA changes in cartilage, osteophyte formation as well as synovial inflammation got inhibited in early OA progression.

The authors present a well-written manuscript. They have investigated the knee joint component changes in histomorphological level such as knee cartilage, osteophyte and synovium. The manuscript is of interest to the readership of PlosOne and should be published after the following minor points have been addressed and corrected by authors:

1. Title: Reduction of knee joint load suppresses cartilage degeneration, osteophyte formation, and synovitis in early stage using a post-traumatic rat model. Missing a subject for early stage, consider adding “early stage of osteoarthritis” or consider change title to “Reduction of knee joint load alleviates the early stage of osteoarthritis progression by suppressing cartilage degeneration, osteophyte formation, and synovitis using a post traumatic rat model”.

2. Line 28: The study aim was… consider change to “The aim of this study was” or “The study’s aim was”.

3. Line 109: 2 weeks and 4 weeks. You have mentioned the reason why to study 2 weeks and 4 weeks later in your discussion, it would be better to specify it here as well.

4. Line 146: Three paraffin sections spanning across the entire knee joint? Please explain or add a reference of why three paraffin sections were chosen.

5. Line 208: All data were analyzed as parametric data? Please identify the data are normally distributed or not.

6. Line 287: “Histological effect of reduced knee joint loading to on osteophyte”, please delete “to”.

7. Line 329: “euthanasia by dropout from the experimental design”. It should be “dropout by euthanasia from the experimental design”.

8. For the body weight, there is a significant difference between OAHS group and OA group at 2 weeks. Can you give some speculations of why the body weight drops at 2 weeks in OAHS group?

9. Figure 4B, y-axis label incorrect. The graph data are not consistent with S5 Table Histomorphometrical results-Matrix intensity data.

10. In the figure, the operated group showing as DMM. In manuscript and supplementary document, the operated group showing as operated. Please keep consistent by using either DMM or operated.

11. Ref: some references are without doi, formats are not consistent.

Reviewer #2: The paper investigated on the effect of reduced load on a DMM-induced OA progression. The topic is of great interest and the results show promising evidence that the reduction of knee joint load supressed the OA progression. However I have 2 major concerns.

1. As the author mentioned, the OA group at 2 and 4 weeks data are from the previous study. However I did not see any explanation about how the surgical procedure was controlled to be consistent between the previous study and this study (I assume the surgeries are done at different times and/or by different personnel?). Only staining method control was described. DMM surgeries performed by different personnels can result in different outcomes due to operational variations so it is important the author have this addressed.

2. Also mentioned in the discussion section, the study used only histological and histomorphometrical analysis to support the conclusion. This does not seem scientifically justifiable. There are several studies on the mechanical properties of articular cartilage and it does not always go consistent with histological signs. For example in Doyran et al 2017 Osteoarthritis Cartilage, it was demonstrated that after the DMM surgery, mechanical properties start to decrease just a few days after the surgery while histological signs doesn't show up until 2-4 weeks after the surgery. The reduced histological sign suggests that the DMM-induced OA is delayed by reduced load but at least one more evidence is needed to demonstrate the results, for example immunohistochemistry staining, biochemical assay, or mechanical test.

6. PLOS authors have the option to publish the peer review history of their article (what does this mean?). If published, this will include your full peer review and any attached files.

Reviewer #1: No

Reviewer #2: No

---

## [Author Response · Author response to Decision Letter 0]

11 May 2021

Responses to Journal Editor’s Additional Comments

Comment 1

As part of your revisions, PLOS ONE asked to provide additional methodological details pertaining to animal research, health and well-being. In response to this request, you have kindly revised your manuscript to include the following details (lines 101 - 103):

"After starting the experiment, no further interventions, including range of motion exercise, were performed during the experimental period. No analgesics or anti-inflammatory drugs were administered to any of the rats."

Can you please clarify - did you provide animals with peri- or post-operative analgesics? If you did not, please provide a scientific justification and indicate whether your animal research oversight committee/IACUC specifically approved this failure to provide pain relief.

Author Action:

In accordance with the editor’s comment, we have revised the following sentences.

Original sentence:

No analgesics or anti-inflammatory drugs were administered to any of the rats.

Revised sentences (page 6, lines 102–105):

No analgesics or anti-inflammatory drugs were administered to any of the rats during the peri-operative period. The non-use of these drugs was approved by the Animal Research Committee of our university as described above and was following previous studies [3,12,22,23].

Comment 2

Additionally, you write the following (lines 141-142):"Since our previous study [12] is open-access and copyrighted under a CC-BY license, reuse of the data is permitted." Please replace the word, "copyrighted" with "published".

(From https://creativecommons.org/about/program-areas/open-access/: "Open access literature is defined as 'digital, online, free of charge, and free of most copyright and licensing restrictions. The recommendations of the Budapest Open Access Declaration—including the use of liberal licensing (such as CC BY)— is widely recognized in the community as a means to make a work truly open access.")

Author Action:

In accordance with the editor’s comment, we have replaced the word "copyrighted" with "published" in the corresponding sentence.

 

Responses to Academic Editor’s Comments

Comment 1

Please ensure that your manuscript meets PLOS ONE's style requirements, including those for file naming. The PLOS ONE style templates can be found at https://journals.plos.org/plosone/s/file?id=wjVg/PLOSOne_formatting_sample_main_body.pdf and https://journals.plos.org/plosone/s/file?id=ba62/PLOSOne_formatting_sample_title_authors_affiliations.pdf

Author Action:

We have revised the manuscript, file naming, and author information in accordance with the journal’s style requirements.

Comment 2

Thank you for submitting the above manuscript to PLOS ONE. During our internal evaluation of the manuscript, we found significant text overlap between your submission and the following previously published works, some of which you are an author.

https://journals.sagepub.com/doi/10.1177/1947603520982350

https://www.oarsijournal.com/article/S1063-4584(19)30896-9/fulltext

Please revise the manuscript to rephrase the duplicated text, cite your sources, and provide details as to how the current manuscript advances on previous work. Please note that further consideration is dependent on the submission of a manuscript that addresses these concerns about the overlap in text with published work.

Author Action:

We completely agree with the editor’s comment. As the editor pointed out, we identified sections that were identical or similar to our previous study. We have revised all the relevant sections, aside from the Methods section. However, if by any chance, identical or similar phrasing remains, please let us know; we will do our best to correct it immediately.

Comment 3

As part of your revisions, please update your Methods/Results to address the following items pertaining to animal research, animal health and welfare. (1) The frequency of animal monitoring, including the specific criteria you used to monitor animal health as well as a description of humane endpoints. (2) Specific animal welfare considerations, including efforts to alleviate suffering (supportive care - gel-packs, cage modifications/environmental enrichment, administration of analgesics, anesthetics, etc.), (3) The rate of mortality during the study (if applicable) and if animals died unexpectedly or prior to the experimental endpoint, an explanation for the cause of death. (4) Details of unanticipated adverse events such as illness or injury as a result of the experimental procedures; (5) Lastly, please complete and submit the ARRIVE Guidelines 2.0 checklist (Essential 10 version): https://arriveguidelines.org/resources/author-checklists.

Author Action:

In accordance with the reviewer’s comment, we have added the following sentences. We have updated and resubmitted the ARRIVE Guidelines 2.0 checklist (Essential 10 version) to be based on the revisions.

Added information (page 5–6, lines 86–89):

The experimental animals were monitored 2–3 times per week to control their health status, including general food and water intake, surgical wound condition, gait, and hindlimb suspension. The experimenter cleaned the cages once or twice every 2 weeks to keep the breeding environment clean.

Added information (page 13, lines 222–225):

No experimental animals died through accidents or other unexpected causes during any of the experiments. None of the rats developed infections in their knee joints or tail wounds. All animals completed the study safely, as per the experimental design, without any unexpected adverse events.

 

Responses to Reviewer #1 Comments

Comment 1: 

Title: Reduction of knee joint load suppresses cartilage degeneration, osteophyte formation, and synovitis in early stage using a post-traumatic rat model. Missing a subject for early stage, consider adding “early stage of osteoarthritis” or consider change title to “Reduction of knee joint load alleviates the early stage of osteoarthritis progression by suppressing cartilage degeneration, osteophyte formation, and synovitis using a post traumatic rat model”.

Author Action:

In accordance with the reviewer’s comment, we have revised the title.

Original title:

Reduction of knee joint load suppresses cartilage degeneration, osteophyte formation, and synovitis in early stage using a post-traumatic rat model

Revised title:

Reduction of knee joint load suppresses cartilage degeneration, osteophyte formation, and synovitis in early-stage osteoarthritis using a post-traumatic rat model

Comment 2: 

Line 28: The study aim was… consider change to “The aim of this study was” or “The study’s aim was”.

Author Action:

In response to the reviewer’s comment, we have revised the sentence.

Original sentence:

The study aim was to clarify the histological effect of reducing the loading to knee on cartilage degeneration, osteophyte formation, and synovitis as osteoarthritis (OA) progresses in the early stage using a post-traumatic rat model.

Revised sentence (page 2, lines 17–19):

The purpose of this study was to clarify the histological effect of reducing the loading to knee on cartilage degeneration, osteophyte formation, and synovitis in early-stage osteoarthritis (OA) using a post-traumatic rat model.

Comment 3: 

Line 109: 2 weeks and 4 weeks. You have mentioned the reason why to study 2 weeks and 4 weeks later in your discussion, it would be better to specify it here as well.

Author Action:

In accordance with the reviewer’s comment, we have added the following sentences.

Added sentences (page 6, lines 96–100):

Although it would have been ideal to evaluate OA until the end stage, long-term experimentation was difficult in the case of hindlimb suspension. Long-term suspension increases the suffering of the experimental animals and in some cases requires euthanasia. This also results in an increase in the number of experimental animals required. Consequently, this study focused only on early-stage OA.

Comment 4: 

Line 146: Three paraffin sections spanning across the entire knee joint? Please explain or add a reference of why three paraffin sections were chosen.

Author Action:

We evaluated the articular cartilage of the center of the tibia in the medial tibiofemoral joint. We chose three paraffin sections to assess the extent of the osteoarthritis lesions, as shown in lines 166–169 of the manuscript. This method has been reported in previous studies by Iijima et al1, 2. These two references have been added as citations. In the present study, we used the same method as in the previous studies to evaluate histological changes. Specifically, three paraffin sections were created every 200 μm, and the total OARSI score was calculated by summing the OARSI scores of the three sections to evaluate the extent of the lesion. Also, the total OARSI score of the three paraffin sections was used as the total OARSI score to evaluate the severity of osteoarthritis.

Following the reviewer’s comment, we have revised the below-mentioned sentences. Additionally, the following two references have been added as citations.

Original sentence:

As described previously [12], decalcified paraffin sections were prepared for histology. Both knees were excised frontally to evaluate the histological changes in the medial tibiofemoral joints [11,13]. Three paraffin sections (3-µm thickness) spaced at 200-µm intervals spanning the entire knee joint were stained with hematoxylin–eosin and toluidine blue to evaluate the severity of cartilage lesions.

Revised sentence (page 8, lines 137–141):

As described previously [12], decalcified paraffin sections were prepared for histology. Both knees were excised frontally to evaluate the histological changes in the medial tibiofemoral joints [11,13]. Three paraffin sections (3-µm thickness) spaced at 200-µm intervals spanning the center of the medial tibiofemoral joint were stained with hematoxylin–eosin and toluidine blue to evaluate the severity and extent of cartilage lesions [26,27].

1. Iijima H, Aoyama T, Ito A, Tajino J, Yamaguchi S, Nagai M, et al. Exercise intervention increases expression of bone morphogenetic proteins and prevents the progression of cartilage-subchondral bone lesions in a post-traumatic rat knee model. Osteoarthr Cartil. 2016;24: 1092-1102. doi:10.1016/j.joca.2016.01.006.

2. Iijima H, Aoyama T, Ito A, Tajino J, Yamaguchi S, Nagai M, et al. Physiological exercise loading suppresses post-traumatic osteoarthritis progression via an increase in bone morphogenetic proteins expression in an experimental rat knee model. Osteoarthr Cartil. 2017;25: 964-975. doi:10.1016/j.joca.2016.12.008.

Comment 5: 

Line 208: All data were analyzed as parametric data? Please identify the data are normally distributed or not.

Author Action:

Thank you for pointing this out. We had received the critique previously, in the peer review of our submission1. In that case, the reviewer, presumed to be a biostatistician, had instructed us as follows: “With sample size of 5 per group, it is not possible to assess the distribution of the data from the data itself and use of tests, such as Shapiro–Wilk or Levene is misleading. The authors should consider what is known about the distribution of weight, OARSI and Mankin scores and choose a statistical model thereafter. I would recommend using parametric methods.” 

In addition, at that time, because it was difficult for us to accurately resolve this reviewer’s comment, we consulted with a Professor and the Chief of the Department of Biostatistics at the Innovative Clinical Research Center, Kanazawa University Hospital, who suggested using parametric methods, similar to the advice of the peer reviewer. 

Therefore, we did not test the normality and homogeneity of variance between each group because of the small sample size and used Tukey’s Honest Significant Difference test for all groups. 

Thus, in the present study, we used a parametric method because we performed the study with a small sample size. In addition, we used G*power to ensure statistical sample size validity.

1. Takahashi I, Matsuzaki T, Kuroki H, Hoso M. Joint unloading inhibits articular cartilage degeneration in knee joints of a monosodium iodoacetate-induced rat model of osteoarthritis. Osteoarthr Cartil. 2019;27: 1084-1093. doi:10.1016/j.joca.2019.03.001.

Comment 6: 

Line 287: “Histological effect of reduced knee joint loading to on osteophyte”, please delete “to”.

Author Action:

According to the reviewer’s comment, we have revised the following sentence.

Original sentence:

Figure 5. Histological effect of reduced knee joint loading to on osteophyte formation of the medial tibia.

Revised sentence (page 16, lines 282):

Fig 5. Results of histological and histomorphological changes in osteophyte formation.

Comment 7: 

Line 329: “euthanasia by dropout from the experimental design”. It should be “dropout by euthanasia from the experimental design”.

Author Action:

According to the reviewer’s comment, we have revised the following sentence.

Original sentence:

For long-term experiment in tail suspension, tail injury may cause distress to the experimental animal and euthanasia by dropout from the experimental design may be necessary.

Revised sentence (page 19, lines 323–325): 

For long-term experiments in tail suspension, tail injury may cause distress to the experimental animal and dropout by euthanasia from the experimental design may be necessary.

Comment 8: 

For the body weight, there is a significant difference between OAHS group and OA group at 2 weeks. Can you give some speculations of why the body weight drops at 2 weeks in OAHS group?

Author Action:

There are several possible causes for significant weight loss in the OAHS group at 2 weeks. We believe that one factor is stress associated with environmental changes, leading to an associated decrease in food intake and water consumption. Therefore, we believe that the significant weight loss observed at 2 weeks after hindlimb suspension disappeared at 4 weeks after hindlimb suspension due to adaptation to the rearing environment. However, we did not measure the food and water consumption of the animals or the serum albumin content and total protein level, so this remains speculative. To the best of our knowledge, there are some previous studies on hindlimb suspension using rats. In previous studies that described changes in body weight, there is currently no consensus on the cause of these weight changes. Some studies have reported that body weight of hindlimb-suspension rats was not significantly different than that of the control rats1-3. Conversely, other studies have reported that body weight of the hindlimb-suspension rats was lower than that of the control rats4,5. 

1. Matsuzaki T, Yosida S, Ikeda A, Hoso M. Changes in joint components after knee immobilization associated with hindlimb unweighting in rats. Journal of Wellness and Health Care 2018;42:33-40.

2. Kaneguchi A, Ozawa J, Kawamata S, Kurose T, Yamaoka K. Intermittent whole-body vibration attenuates a reduction in the number of the capillaries in unloaded rat skeletal muscle. BMC Musculoskeltal Disorders 2014;15:315.

3. Ferreira JA, Crissey JM, Brown M. An alternant method to the traditional NASA hindlimb unloading model in mice. Journal of Visualized Experiments 2011;(49):e2467.

4. O'Connor KM. Unweighting accelerates tidemark advancement in articular cartilage at the knee joint of rats. Journal of bone and mineral research: the official journal of the American Society for Bone and Mineral Research 1997;12:580-9.

5. Takahashi I, Matsuzaki T, Kuroki H, Hoso M. Joint unloading inhibits articular cartilage degeneration in knee joints of a monosodium iodoacetate-induced rat model of osteoarthritis. Osteoarthritis and Cartilage 2019;27:1084-93.

Comment 9: 

Figure 4B, y-axis label incorrect. The graph data are not consistent with S5 Table Histomorphometrical results-Matrix intensity data.

Author Action:

We thank the reviewer for pointing this out, but we would like to defend the current presentation of the graph. We intentionally reversed the y-axis from 0-255. The reason for this is that darker staining is normal for toluidine blue staining, while lighter staining is abnormal, such as in osteoarthritis. However, in the data, darker staining intensity is calculated as the lower value and lighter staining intensity as the higher value. In other words, if the graph is represented without inverting the y-axis, specimens stained darker, closer to normal, will show lower values, which may be mistakenly understood as worsening. Therefore, we inverted the y-axis to enhance visual understanding of the graph.

Comment 10: 

In the figure, the operated group showing as DMM. In manuscript and supplementary document, the operated group showing as operated. Please keep consistent by using either DMM or operated.

Author Action:

In response to the reviewer’s comment, we have standardized our terminology to “operated.”

Comment 11: 

Ref: some references are without doi, formats are not consistent.

Author Action:

We have revised the reference format according to the journal’s style requirements.

 

Responses to Reviewer #2 Comments

Comment 1: 

As the author mentioned, the OA group at 2 and 4 weeks data are from the previous study. However I did not see any explanation about how the surgical procedure was controlled to be consistent between the previous study and this study (I assume the surgeries are done at different times and/or by different personnel?). Only staining method control was described. DMM surgeries performed by different personnels can result in different outcomes due to operational variations so it is important the author have this addressed.

Author Action:

We completely agree with the reviewer’s comment. As pointed out by the reviewer, we created the OA and OAHS groups in two phases. Therefore, we have made efforts to standardize the surgery and staining techniques. Specifically, throughout the two surgeries, we used the same experimental technique. Further, the same two surgeons conducted the experiments using identical protocols and techniques to create OA rats by DMM. In addition, we confirmed their proficiency in the surgical technique via preliminary experiments. Furthermore, when the surgical time was measured, no obvious difference was observed between the two experiments. As for the experimental technique, we followed that of a previous study1. Hence, we believe that we have ensured high reproducibility and accuracy in the two surgical procedures.

1. Glasson SS, Blanchet TJ, Morris EA. The surgical destabilization of the medial meniscus (DMM) model of osteoarthritis in the 129/SvEv mouse. Osteoarthr Cartil. 2007;15: 1061–1069. doi:10.1016/j.joca.2007.03.006.

Accordingly, we have revised and added a description on this in the Discussion as follows:

Original sentences

In the present study, we conducted additional experiments after the primary one [12]. To avoid any obvious differences in staining between the two experiments, we made four technical checks. First, we used the same staining conditions and environmental factors (laboratory, dyeing solution, room temperature, and used goods). Second, before conducting the experiments, we confirmed the absence of apparent differences in the staining intensity. Third, in the histological evaluation, the evaluator was blinded to all experimental conditions, including group assignment and the experimental period. Fourth, the intra- and inter-rater reliabilities were very high [13]. Therefore, it was possible to perform highly scientific experiments with assurance of objectivity and reproducibility.

Revised sentences (pages 20, lines 340-353):

 The present study consists of the results of our previous study and additional experiments specified herein [12]. To avoid significant differences in experimental techniques between the two experiments, we paid careful attention to three aspects: surgery, staining, and evaluation. OA models were created by the same surgeon using the same surgical technique as in the previous study [22,23]. Additionally, the surgeon’s proficiency in the surgical technique was confirmed by preliminary experimentation. Furthermore, there was no marked difference in operative time between experiments. Staining was performed in the same environment and conditions, including staining solution used, staining time, room temperature, and laboratory equipment. Before conducting additional experiments, we preliminarily verified that there was no marked difference in staining intensity by the same technique. Histological evaluation was performed by two well-trained double-blinded experimenters. The same experimenter performed the histological evaluation in the two experiments and the two had high intra- and inter-rater reliability [13]. Based on the above, we consider our two experiments to have been conducted with high reliability and reproducibility.

Comment 2: 

Also mentioned in the discussion section, the study used only histological and histomorphometrical analysis to support the conclusion. This does not seem scientifically justifiable. There are several studies on the mechanical properties of articular cartilage and it does not always go consistent with histological signs. For example in Doyran et al 2017 Osteoarthritis Cartilage, it was demonstrated that after the DMM surgery, mechanical properties start to decrease just a few days after the surgery while histological signs doesn't show up until 2-4 weeks after the surgery. The reduced histological sign suggests that the DMM-induced OA is delayed by reduced load but at least one more evidence is needed to demonstrate the results, for example immunohistochemistry staining, biochemical assay, or mechanical test.

Author Action:

We completely understand the concern mentioned in the reviewer’s comment. Ideally, we also believe that multiple analytical methods should be performed. However, in terms of study period, funding, and experimental equipment, we were unable to conduct any additional experiments using these analytical methods. Fortunately, we were able to obtain novel results through histological and histomorphometrical analyses alone. Therefore, we have added a description of study limitations to the Discussion and revised the concluding statement of the Discussion as follows: 

Revised description 

Original sentences

First, only histological and histomorphometrical analysis method was performed. In order to clarify the mechanism of the effect of load reduction on OA progression, further studies for catabolic markers, such as MMP-13 and ADAMTS5, or anabolic markers, such as aggrecan and type II collagen, would be necessary using immunohistochemical staining, situ hybridization, western blotting, and polymerase chain reaction.

Revised sentences (page 21, lines 354–358):

First, only histological and histomorphometrical analyses were performed. To clarify the effect of load reduction on OA progression, further biochemical studies for catabolic markers, such as MMP-13 and ADAMTS5, or anabolic markers, such as aggrecan and type II collagen, would be necessary, and further studies using mechanical tests would also be needed.

Revised description

Original sentences

Further studies using biochemical analyses to examine protein and gene expression are required to elucidate the mechanism underlying delayed OA progression caused by joint-load reduction.

Revised sentences (pages 22, lines 377–378):

Further studies using biochemical and mechanical analyses are required to clarify the structure underlying delayed OA progression caused by joint-load reduction.

---

## [Decision Letter · Decision Letter 1]

25 Jun 2021

Reduction of knee joint load suppresses cartilage degeneration, osteophyte formation, and synovitis in early-stage osteoarthritis using a post-traumatic rat model

PONE-D-21-06339R1

Dear Dr. Takahashi,

We’re pleased to inform you that your manuscript has been judged scientifically suitable for publication and will be formally accepted for publication once it meets all outstanding technical requirements.

Kind regards,

Lin Han

Academic Editor

PLOS ONE

Additional Editor Comments (optional):

Reviewers' comments:

Reviewer's Responses to Questions

**Comments to the Author**

1. If the authors have adequately addressed your comments raised in a previous round of review and you feel that this manuscript is now acceptable for publication, you may indicate that here to bypass the “Comments to the Author” section, enter your conflict of interest statement in the “Confidential to Editor” section, and submit your "Accept" recommendation.

Reviewer #1: All comments have been addressed

Reviewer #2: All comments have been addressed

2. Is the manuscript technically sound, and do the data support the conclusions?

Reviewer #1: (No Response)

Reviewer #2: Yes

3. Has the statistical analysis been performed appropriately and rigorously? 

Reviewer #1: (No Response)

Reviewer #2: Yes

4. Have the authors made all data underlying the findings in their manuscript fully available?

Reviewer #1: (No Response)

Reviewer #2: Yes

5. Is the manuscript presented in an intelligible fashion and written in standard English?

Reviewer #1: (No Response)

Reviewer #2: Yes

6. Review Comments to the Author

Reviewer #1: (No Response)

Reviewer #2: All my previous comments have been properly addressed. Thank you for the great opportunity to review this paper and great work!

7. PLOS authors have the option to publish the peer review history of their article (what does this mean?). If published, this will include your full peer review and any attached files.

Reviewer #1: No

Reviewer #2: No

---

## [Editor Report · Acceptance letter]

7 Jul 2021

PONE-D-21-06339R1 

Reduction of knee joint load suppresses cartilage degeneration, osteophyte formation, and synovitis in early-stage osteoarthritis using a post-traumatic rat model 

Dear Dr. Takahashi:

I'm pleased to inform you that your manuscript has been deemed suitable for publication in PLOS ONE. Congratulations! Your manuscript is now with our production department. 

Kind regards, 

on behalf of

Dr. Lin Han 

Academic Editor

PLOS ONE